# Routine Ophthalmological Examination Rates in Adults with Sickle Cell Disease Are Low and Must Be Improved

**DOI:** 10.3390/ijerph20043451

**Published:** 2023-02-16

**Authors:** Patricia Zulueta, Caterina P. Minniti, Anvit Rai, Tiana J. Toribio, Jee-Young Moon, Umar K. Mian

**Affiliations:** 1Albert Einstein College of Medicine, New York, NY 10461, USA; 2Department of Ophthalmology & Visual Sciences, Montefiore Medical Center, Albert Einstein College of Medicine, New York, NY 10461, USA; 3Department of Pediatrics, Montefiore Medical Center, Albert Einstein College of Medicine, New York, NY 10461, USA; 4Department of Epidemiology & Population Health, Montefiore Medical Center, Albert Einstein College of Medicine, New York, NY 10461, USA

**Keywords:** sickle cell disease, sickle cell retinopathy, screening

## Abstract

The American Academy of Ophthalmology and the National Heart, Lung and Blood Institute recommend patients with sickle cell disease (SCD) undergo dilated funduscopic exams (DFE) every 1–2 years to screen for sickle retinopathy. There is a paucity of data on the adherence rate to these guidelines; a retrospective study was performed to evaluate our institution’s adherence. A chart review of 842 adults with SCD, seen 3/2017–3/2021 in the Montefiore healthcare system (*All Patients*), was done. Only about half of *All Patients* (n = 842) had >1 DFE during the study period (*Total Examined Patients*, n = 415). The *Total Examined Patients* were categorized as screening, those without retinopathy (*Retinopathy−*, n = 199), or follow-up, including individuals previously diagnosed with retinopathy (*Retinopathy+*, n = 216). Only 40.3% of screening patients (n = 87) had DFE at least biennially. As expected, there was a significant decrease in the average DFE rate of the *Total Examined Patients* after the COVID-19 pandemic started (13.6%) compared to pre-COVID (29.8%, *p <* 0.001). Similarly, there was a significant decrease in the screening rate of *Retinopathy−* patients from 18.6% on average pre-COVID to 6.7% during COVID (*p <* 0.001). This data shows the sickle retinopathy screening rate is low and innovative approaches may need to be employed to remedy this issue.

## 1. Introduction

Sickle cell disease (SCD) is a genetic defect of hemoglobin causing RBCs to sickle upon deoxygenation. Sickled RBCs, ongoing hemolysis and hypoxia cause endothelial damage to blood vessels, including the retinal vessels [1]. The ischemic-reperfusion injury (I/RI) that results from these changes at the molecular level is the basis for the vasculopathies associated with SCD. I/RI can be divided into two phases, the initial ischemic phase followed by a reperfusion state. The reperfusion state further promotes the endothelial pro-inflammatory state and leads to reperfusion injury in the tissue [1].

Chronic cycling of I/RI in the retina releases vascular endothelial growth factor (VEGF), leading to sickle cell retinopathy (SCR), which can be non-proliferative, without neovascularization or proliferative retinopathy (PSR) with neovascularization. VEGF promotes the growth of abnormal new vasculature (PSR), which is prone to bleeding, causing vitreous hemorrhage or contraction, creating tractional retinal detachments [2]. Vitreous hemorrhage, tractional retinal detachment, and other proliferative changes in the eye may cause visual loss in up to 10–20% of patients with SCD [3].

The two most commonly sickle cell genotypes are HbSS and HbSC, which present with different rates of ophthalmic complications. Per the AAO, PSR affects up to 40% of patients with the HbSC genotype and up to 20% of patients with the HbSS genotype. One theory posits that the HbSC genotype causes greater blood viscosity, leading to occlusion in the retinal vasculature and predisposing these patients to higher rates of PSR as compared to patients with HbSS. Furthermore, 6% of HbSS and 32% of HbSC patients will develop vitreous hemorrhages, which may require a vitrectomy [4]. As such, it is possible that optimal ophthalmic screening and management may differ between the two predominant sickle cell genotypes. Other forms of sickle cell disease include HbS beta thalassemia and other rare variants [2]. Interestingly, concerning patient awareness of sickle cell disease complications, a survey by Alsheri et al. found that the majority (57.3%) of their SCD patients were not aware of the ocular complications of SCD, which may affect the number of patients seeking routine eye examinations [5].

Evidence on the treatment of PSR is scarce, whether it is observation, self-regression, laserpexy to decrease retinal oxygen demand and prevent further ischemia or intravitreal anti-VEGF to temporarily block VEGF [6]. Currently, there are no large randomized clinical trials that have looked at the efficacy of anti-VEGF injections in SCR. However, a survey sent to 95 retinal specialists found that roughly 48% do not use anti-VEGF injections. Among the 52% that did use these injections, most used them after a patient had developed vitreous hemorrhages [7]. A Cochran database systematic review (2015) found only two previous trials that assessed the efficacy of laser photocoagulation. Laser treatment did not significantly affect the regression of sickle cell retinopathy, nor did it prevent the progression of retinopathy. However, laser photocoagulation averts visual loss and vitreous hemorrhages [8]. As such, screening SCD patients for retinopathy and subsequently treating patients with early signs of PSR may prevent severe ocular morbidity.

Screening recommendations by the American Academy of Ophthalmology (AAO) and the National Heart, Lung, and Blood Institute (NHLBI, 2014) are similar. Both organizations suggest sickle cell patients receive routine dilated funduscopic exams (DFE) to screen for retinopathy every 1–2 years, beginning at age 10 [1,9]. However, the NHLBI report indicated they could not find data evaluating screening intervals or the diagnostic accuracy of SCR screening [9]. Thus, both the AAO and NHLBI guidelines are not based on actual clinical trials but on expert opinion and consensus. While these societies recommend the use of DFE, other authors have proposed annual screening with techniques including optical coherence tomography and ultra-wide-field fluorescein angiography [10].

SCR screening guidelines have been considered in the pediatric population. Li et al. performed a retrospective cohort study of 398 children with SCD seen by an ophthalmologist over 4 years at the Children’s Hospital of Philadelphia. The included subjects had a mean number of 2.2 ophthalmology visits (minimum 1.0, maximum 15.0) during the study period, which suggests many of these patients were screened at least biennially. Based on their eye examination data, the authors recommend screening at 9 and 13 years old for patients with the SC and SS genotypes, respectively. The mean age at the first ophthalmology visit in their study was 9.6 years, with a range from 0.0 to 18.0 years—the percentage of children screened appropriately prior to 9 and 13 years old for SC and SS genotype, respectively, was not provided [11].

While screening in children has been examined, there is a paucity of recent data reporting whether the AAO and NHLBI guidelines are followed in practice in the adult SCD population. The purpose of this study is to assess the adherence to the AAO and NHLBI screening recommendations in adults with SCD at a large urban academic center.

## 2. Materials and Methods

This study is a retrospective chart review from 3/2017–3/2021 of a population of patients (*All Patients*) with confirmed HbSS or Hb SC diagnosis, identified from a database of SCD patients over the age of 18 having at least one medical encounter with a Montefiore medical physician (General medicine or Hematology). Other sickle cell variants (such as HbS beta thalassemia) were not included in this study due to the low number of patients with these genotypes currently in the SCD database. The inclusion criteria were patients with a DFE during the study period (*Total Examined Patients*). The *Total Examined Patients* were divided into two groups: adults who were screened by Montefiore Ophthalmology and found not to have retinopathy previously (*Retinopathy−*) and patients that were already identified by Montefiore Ophthalmology as having an abnormal retinal exam with a follow-up DFE at our institution (*Retinopathy+*). Patients with a diagnosis of an abnormal retinal exam from an outside institution were not included in this study. *Retinopathy−* patients are defined as having no SCR or any other retinal conditions that would require more frequent ophthalmology follow-up. Eye exams of *Retinopathy−* patients are also referred to as screening exams and screening DFEs.

Institutional Review Board (IRB) approval from the Albert Einstein College of Medicine was obtained. Data were collected from patients’ Electronic Medical records (EMR, EPIC) for demographic information, SCD genotype, ophthalmological exam and retinal angiogram. A DFE is considered an eye exam in this study. A routine exam was defined as at least one DFE every 2 years. If an angiogram was performed after a DFE, then the initial DFE, the angiogram procedural visit, and the follow-up visit to discuss the angiogram results were all counted as one eye exam.

Rates of screening exams of *Retinopathy−* patients were calculated by removing patients with known SCR from all SCD patients (*All Patients*), as these patients no longer fit the AAO and NHLBI recommendations due to having already been diagnosed with SCR. Rates of *Retinopathy+* eye exams were calculated out of all retinopathic patients seen by a Montefiore medical physician (i.e., *Retinopathy+* and patients with a diagnosis of retinopathy in their chart from an outside institution).

Yearly DFE rates were calculated from March to March since COVID-19 was declared a global pandemic in 3/2020. 3/2020–3/2021 was divided into two six-month periods to more closely track the effect of the COVID-19 pandemic on ophthalmology examination rates.

### Statistical Analysis

Two-tailed Welch’s *t*-test and two-tailed Mann–Whitney U test were both used for statistical analysis of continuous data, Fisher’s exact test for categorical data, two-sample proportion test for comparison of two exam rates and Cochran-Armitage test for trend of exam rates.

## 3. Results

### 3.1. Total Examined Patients

Of the 842 unique patients with SCD with at least one medical encounter during the study period (*All Patients*), 415 had a DFE and constituted the *Total Examined Patients*. 301 (72.5%) of the *Total Examined Patients* had the SS genotype, and 114 patients had the SC genotype. Two hundred sixteen patients with DFEs were found to be *Retinopathy−* (52.0%), and 199 were *Retinopathy+* (48.0%). The *Retinopathy−* group contained significantly less SC genotype as compared to *Retinopathy+*. 31.6% of the *Retinopathy−* patients were SC, whereas 68.5% of the *Retinopathy+* patients were SC (*p <* 0.001; Table 1).

The average age of the *Total Examined Patients* was 37.5 years, but it was significantly younger for the *Retinopathy−* patients compared to *Retinopathy+* at 33.3 years vs. 42.0 years, respectively (*p <* 0.001). There was no significant difference in gender ratio between the *Retinopathy−* and + groups (Table 1).

The average rate of *Total Examined Patients* with DFE (s) pre-COVID, calculated from the mean of the 3 yearly annual rates from 3/2017–3/2020, was low at 29.8%. The yearly *Total Examined Patients*’ DFE rate significantly increased every year, starting at 27.2% for the first year and reaching 33.9% for the last pre-COVID year (*p* = 0.006). As expected, there was a significant and dramatic decrease in the average *Total Examined Patients*’ DFE rate during the pandemic year from 29.8% to 13.6% (3/2020–3/2021; *p <* 0.001), but it is interesting to note that the rate increased nearly three times between the first and the last six months of the COVID year from 7.2% to 20.0% (Table 2, Figure A1).

### 3.2. Retinopathy−

Less than half (n = 87 or 40.3%) of the *Retinopathy−* patients had routine DFEs (i.e., DFEs at least once every 2 years). While there were more females in the *Total Examined Patients*, *Retinopathy+* and *Retinopathy−* groups, there was no statistical difference in sex between routine and non-routine *Retinopathy−* screening (*p* = 0.68). As far as the SS:SC genotype ratio was concerned, there were slightly more SC patients in the routine screening patient group (52.8%) compared to the non-routine screening group (47.2%), but the difference was not significant (*p* = 0.098; Table 3).

We saw very similar trends for the *Retinopathy−* DFE rates as compared to the *Total Examined Patients*’ DFE rates. The average *Retinopathy−* screening rate for the study period prior to COVID was 18.6%, although it increased from 3/2017–3/2018 (16.4%) to 3/2019–3/2020 (22.1%; *p* = 0.014). As expected, the screening rate significantly dropped during the COVID year to an average of 6.7% (*p <* 0.001; Table 2).

### 3.3. Retinopathy+

We observed a similar trend in the follow-up of *Retinopathy+* patients as we did with the *Total Examined Patients*. Throughout the study period, nearly half of the *Total Examined Patients* with eye exams were categorized as *Retinopathy+*. There was a non-significant increase in the pre-COVID years of *Retinopathy+* DFE rate; 41.7% from 3/2017–3/2018 to 50.2% from 3/2019–3/2020 (*p* = 0.068). We saw a significant decrease in the *Retinopathy+* DFE rate from an average of 45.1% pre-COVID to an average of 22.0% during COVID (*p <* 0.001), yet there was a significant increase in the rate between the first and last six months of the COVID year (13.3%–30.7%, *p* = 0.0002).

## 4. Discussion

The AAO and NHLBI recommend that beginning at age 10, sickle cell patients should receive dilated funduscopic exams at least annually or at least biennially. Our study provides data that evaluate whether these guidelines for ophthalmological surveillance of adult SCD patients are being followed in everyday clinical practice. We examined ophthalmologists’ ability to adhere to these recommendations over a four-year period from 3/2017–3/2021.

The ocular characteristics of the SCD patient population seen by Montefiore Ophthalmology (*Total Examined Patients*) are similar to what has been previously reported. The higher proportion of SC patients with retinopathy (68.4% of patients with HbSC genotype were *Retinopathy+*) compared to the SS group (40.2% of patients with HbSS genotype were *Retinopathy+*) is similar to Feroze and Azevedo’s observation that a higher percentage of patients with HbSC develop PSR than patients with HbSS [2]. Saidkasimova et al. showed that age greater than 35 years and SC genotype are risk factors for PSR [12]. We saw a similar trend as patients without retinopathy were significantly younger than patients with retinopathy, which is consistent with several studies that have also found that older age is associated with sight-threatening PSR [12,13,14].

We found that only half of *All Patients* had a DFE during the study period, and 52% of them were seen for screening. Of the *Retinopathy−* patients, about 60% had non-routine DFEs, i.e., examinations with a frequency of less than every 2 years, throughout the entire study period. This indicates that, as we follow our patients over a long period of time (4 years), only a minority are being screened for SCR as often as the AAO/NHLBI guidelines recommend. We did not see any significant differences between SS and SC, age, or sex in terms of patients with routine vs. non-routine screening.

Our data demonstrate that rates of screening DFEs for *Retinopathy−* patients are low. Our average screening rate prior to COVID was 18.6% of eligible patients and decreased to 6.7% during the COVID year. Furthermore, *Retinopathy−* screening rates were consistently lower than the DFE rate of *Total Examined Patients* and the DFE rate of *Retinopathy+*. This is because the *Total Examined Patients* group includes both *Retinopathy−* and *Retinopathy+* patients. *Retinopathy+* patients have previously been diagnosed with SCR, so they require more frequent eye exams to monitor their condition, which increases the DFE rate of *Retinopathy+* and of *Total Examined Patients* as compared to the *Retinopathy−* screening rate.

Our screening rate results expand upon previously published data, showing low adherence to screening guidelines [15]. In 1988 Moriarty et al. reported that only 342 patients received more than 1 eye exam separated by at least 3 months over 10 years out of more than 3000 patients with SCD seen in their clinic [3]. Patient surveys regarding eye exams showed a severe lack of patient knowledge of SCR as a sickle cell complication and a low rate of eye exams. 71.3% of patients with SCD interviewed by Alsheri et al. stated that they never had an eye exam, while 61.6% did not have an eye exam in the last 12 months [5]. Mowatt et al. also administered a questionnaire to patients with SCD; only 28% of subjects answered “Yes” to the question “Do you see your eye doctor for regular checkups?”. Only 42% of respondents answered that they had seen an ophthalmologist once a year, and 21% reported never having seen an ophthalmologist [16].

Inadequate patient and provider education, barriers to making and keeping appointments, transportation and time off work could likely play a role in patients not being screened according to the guidelines. Mowatt et al. administered a questionnaire to patients with SCD on the ocular manifestations of SCD and were given a score based on the percentage of questions answered correctly. Higher scores were correlated to more regular ophthalmological follow-ups. In addition, the authors found that employment was significantly associated with regular eye examinations in SCD patients, suggesting financial stability may be related to patient compliance with proper ophthalmic care [16]. A survey by Alsheri et al. found that the majority (57.3%) of their SCD patients were not aware of the ocular complications of SCD [5]. As such, screening rates may be increased through greater patient involvement, support, and education.

Although our screening rates were low, they increased over time. Pre-COVID, Montefiore saw a significant increase in total DFE rates and screening DFE rates, comparing the data from 3/2017–3/2018 to 3/2019–3/2020. We believe that the close collaboration between Montefiore’s Hematology and Ophthalmology departments led to an increase in regular ophthalmic care.

However, from 3/2020–9/2020, our Ophthalmology department began seeing only urgent patients due to the declaration of the global COVID-19 pandemic. Due to COVID-19 limitations, the Ophthalmology department offered limited visits and accommodated fewer routine fundus examinations as compared to pre-pandemic. The lack of ophthalmologist availability for routine care for our patients is reflected in the significantly decreased average DFE rates after 3/2020 (for *Total Examined Patients*, *Retinopathy−* and *Retinopathy+*). After 9/2020, Montefiore Ophthalmology began seeing routine patients again. There was a subsequent increase in screening DFE rates from 2.8% (3/2020–9/2020) to 10.6% (9/2020–3/2021), along with an increase in DFE rates for *Total Examined Patients* and *Retinopathy+*. Thus, although, as expected, the pandemic negatively impacted the screening of SCD patients and follow-up of retinopathic patients, there was a reasonably quick increase in DFE rates after the first six months of COVID.

Our study has all the limitations that would be associated with a retrospective chart review. Patients receiving ophthalmic care outside our institution were not included, possibly, decreasing the real screening rate. There were 40 such patients not included in the study. Had these patients been included, the *Retinopathy+* group would have increased by 10%. Finally, our study is from a single institution.

## 5. Conclusions

Our relatively low screening rates increased with good cooperation between Hematology and Ophthalmology departments. Every attempt should be made to increase collaboration between the two specialties. Emphasis on improving patient education concerning ophthalmological complications might be a critical factor in improving screening rates. Utilization of novel eye examination techniques may also improve adherence to the AAO and NHLBI’s screening recommendations, e.g., routine ultra-widefield fundus photography or fluorescein angiography. Further studies need to be done to get a clearer picture concerning adherence to the guidelines and ways of improving screening rates.

## Figures and Tables

**Table 1 ijerph-20-03451-t001:** Demographic information of patients with DFEs during the study period.

	*Total Examined Patients* (n = 415)	*Retinopathy+* (n = 199)	*Retinopathy−* (n = 216)	*p* Value
Avg, Median Age (years) Range	37.5, 34.0 18–87	42.0, 40.0 18–87	33.3, 29.5 18–73	<0.001 *
Female %	57.8%	59.3%	56.5%	0.62 ^†^
SS	301 (72.5%)	121 (40.2%)	180 (59.8%)	<0.001 ^†^
SC	114 (27.5%)	78 (68.4%)	36 (31.6%)	

*Total Examined Patients*: SCD patients who had a DFE during the study period. *Retinopathy+*: SCD patients with a previous diagnosis of retinopathy by a Montefiore ophthalmologist with follow-up DFE during the study period. *Retinopathy−*: SCD patients without a previous diagnosis of retinopathy with screening DFE during the study period. * *p*-value was obtained by two-tailed Welch’s *t*-test. The *p*-value from the Mann–Whitney U test was <0.001. ^†^ *p*-value was obtained by Fisher’s exact test comparing sex/genotype distribution among *Retinopathy+* and *Retinopathy−*.

**Table 2 ijerph-20-03451-t002:** Rates of *Total Examined Patients* and Screening Exams (*Retinopathy*−).

Year(s)	*All Patients*	*Total Examined Patients*	Rate of *Total Examined Patients*	Screening Exams (*Retinopathy−*)	Rate of *Screening* Exams (*Retinopathy−*) *
3/2017–3/2018	696	189	27.2%	94	16.4%
3/2018–3/2019	699	199	28.4%	99	17.4%
3/2019–3/2020	691	234	33.9%	122	22.1%
3/2020–9/2020	512	37	7.2%	13	2.8%
9/2020–3/2021	493	99	20.0%	44	10.6%

*All Patients*: Total SCD patients seen by a physician during the timeframe. *Total Examined Patients*: SCD patients who had an eye exam during the timeframe. Rate of *Total Examined Patients*: Rate of SCD patients with an eye exam, *Total Examined Patients*/*All Patients.* Screening exams: Number of SCD patients without a previous diagnosis of retinopathy (*Retinopathy−*) who had a (screening) eye exam during the timeframe. Rate of screening exams: a rate of screened patients among SCD patients without a previous diagnosis of retinopathy (*Retinopathy−*). * Rate of screening exams was calculated as follows: (*Retinopathy−* with screening exams)/(*All Patients −* (retinopathic patients)) = (*Retinopathy−* with screening exams)/(*All Patients −* (SCD patients not seen by a Montefiore ophthalmologist but with retinopathy noted in EMR + (*Retinopathy+*))).

**Table 3 ijerph-20-03451-t003:** *Retinopathy− screening* data from 3/2017–3/2021.

	Routine Eye Exams	Non-Routine Eye Exams	*p* Value
Total (n = 216)	87 (40.3%)	129 (59.7%)	
Avg, Median Age (years) Range	32.8, 31.0 18–65	33.6, 29.0 18–73	0.63 *
Female %	58.6%	55.0%	0.68 ^†^
SS (n = 180)	68 (37.8%)	112 (62.2%)	0.098 ^†^
SC (n = 36)	19 (52.8%)	17 (47.2%)	

Routine eye exams are defined as at least one screening DFE every 2 years. Non-routine eye exams are defined as patients with screening DFE only once during the study period or at least once but fewer than every 2 years during the study period. * *p*-value was obtained by two-tailed Welch’s *t*-test. The *p*-value from Mann–Whitney U test was 0.75. ^†^ *p*-value was obtained by Fisher’s exact test comparing sex/genotype distribution among patients with routine screening eye exams and patients with non-routine screening eye exams.

## Data Availability

The datasets used and/or analyzed during the current study are available from the corresponding author upon reasonable request.

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
