# Peer review of "Routine Ophthalmological Examination Rates in Adults with Sickle Cell Disease Are Low and Must Be Improved"

_ijerph, 2023, doi:10.3390/ijerph20043451_

Round 1
Reviewer 1 Report
The manuscript presented the importance of ophthalmological exam rates regarding sickle cell disease. It was well elaborated and delivered, however, it could be improved.
As the results mentioned different genotypes HbSS and HbSC, however, there is no background introduction regarding different genotypes, as well as the information on why we should care about the other genotypes in this research topic.
Secondly, in my understanding, Welch's t-test is more powerful when the data are in normaldistribution. Is the age distribution normal in this case?
Altogether, I suggest the manuscript to be published after minor revision.
Author Response
Point 1: As the results mentioned different genotypes HbSS and HbSC, however, there is no background introduction regarding different genotypes, as well as the information on why we should care about the other genotypes in this research topic.
Response 1: Based on reviewer 1’s feedback, on page 2 from lines 46 – 57 further emphasis was placed on the different rates of ophthalmic complications and possible need for differing management in HbSS vs. HbSC genotypes. The differing presentations of retinopathy in HbSS and HbSC warrant investigation, which is why HbSS and HbSC patients were compared in this paper. In the Materials and Methods section on page 3, lines 97 – 99, we explained that only HbSS and HbSC genotypes were included in this study due to the low number of patients with other sickle cell variants in our database.
Point 2: Secondly, in my understanding, Welch's t-test is more powerful when the data are in normal distribution. Is the age distribution normal in this case?
Response 2: Welch’s t-test was still chosen for statistical analysis of continuous data given the large sample size. However, we also double-checked p-values of age for Retinopathy - vs. Retinopathy + and Routine vs. Non-routine eye exams by Mann-Whitney U test, which does not assume normality. By both the Welch’s t-test and Mann-Whitney U test, we still found that Retinopathy - patients were significantly younger than Retinopathy + and there was no significant difference in age for Non-routine eye exams vs. Routine groups. A section on statistical analysis of age data was added to Appendix A on page 8, lines 344-348.

Reviewer 2 Report
The authors retrospectively reviewed the rates of dilated fundal examination in patients with sickle cell disease at their institution, and overall identified low rates of examination. From a public health standpoint, it is obvious that rates of ophthalmological screening need to be improved, and I believe it would be reasonable to clearly define referral and screening pathways. Out of 842 patients with sickle cell disease, only 415 patients had fundal examination at some point. What were the limiting factors restricting access to ophthalmological screening in the first place (for example, patients not referred at all by a hematologist/physician, versus inability of ophthalmology department to accommodate)? I would invite the authors to kindly elaborate on this further.
Author Response
Point 1: What were the limiting factors restricting access to ophthalmological screening in the first place (for example, patients not referred at all by a hematologist/physician, versus inability of ophthalmology department to accommodate)?
Response 1: We appreciate reviewer 2’s question regarding limiting factors restricting access to ophthalmology screening. We have clarified on page 7, lines 274 – 276 that the Ophthalmology department could not accommodate as many routine fundus exam visits due to the COVID-19 pandemic.
Reviewer 3 Report
The authors describe the rate of routine fundus examination for sickle cell retinopathy in patients with sickle cell disease. The study was also organized, and the exam rate before and after the spread of Covid19 infection was also summarized, which was very useful. It would be even better if the following points were corrected.
Page5 Line 200 The number of references is not parenthesized, causing confusion.
Page7 Line259-263 I think it would be better to describe the Limitation in the Discussion part rather than the Conclusion part.
Author Response
Point 1: Page5 Line 200 The number of references is not parenthesized, causing confusion.
Response 1: We have parenthesized the mentioned reference.
Point 2: Page7 Line259-263 I think it would be better to describe the Limitation in the Discussion part rather than the Conclusion part.
Response 2: We have moved the paragraph regarding study limitations to the Discussion part of the paper (page 7, lines 285 – 289).